# How far can air pollution affect tourism in China? Evidence from panel unconditional quantile regressions

Yuxuan Xiao[1☯], Will W. Qiang[2☯], Chung-Shing Chan[2], Steve H. L. Yim[3], Harry F. Lee[2]*

1 Pingshan Research Center of Planning and Natural Resources in Shenzhen, Shenzhen, China, 2 Department of Geography and Resource Management, The Chinese University of Hong Kong, Shatin, New Territories, Hong Kong, 3 Asian School of the Environment, Nanyang Technological University, Singapore, Singapore

☯ These authors contributed equally to this work.
* harrylee@cuhk.edu.hk

**Data Availability Statement:** All relevant data are within the manuscript and its Supporting Information files.

## Abstract

Previous studies provide empirical evidence for the connection between air pollution and tourism. However, many of them take the nexus as a linear one. It remains unexplored whether any thresholds are required for the nexus to materialize. This study systematically investigates whether PM$_{2.5}$ concentrations–an essential indicator of air pollution–affect tourism in China at various tourism development levels. We analyze 284 Chinese cities from 2008 to 2018 using the Unconditional Quantile Regression method. Our statistical results reveal that air pollution positively influences tourism (regarding tourist visits and tourism revenue) in areas with low tourism development levels. However, a complex correlation between air pollution and tourism emerges when tourism development has reached a certain level. The correlation is initially negative, then positive, and finally disappears. But, the overall correlation remains negative. The effects of the interaction between air pollution and tourism resources on tourism are inverted U-shaped, implying that tourism resources can mitigate the negative effects of air pollution on tourism only when tourism development has reached a certain level. Based on the above findings, the associated policy implications are discussed.

## 1. Introduction

Sustainable tourism development represents the harmonious economy, society, and environment growth, ensuring the tourism industry's long-term resilience [1]. As a critical sector for many countries, tourism promotes income generation and economic growth [2–4]. Nonetheless, the industry's vulnerability to environmental factors and external shocks jeopardizes its future development [5, 6]. Regarding China's tourism development, the tourism industry's total contribution to GDP in 2018 was 11.04%, and direct and indirect employment in the tourism industry accounted for 10.29% of total employment in the country [7]. On the other hand, air pollution has resulted in significant economic losses for China's tourism industry,

**Funding:** Koon Wah Mirror Group, grant no. 6904069, Harry F. LEE Direct Grant for Research of The Chinese University of Hong Kong, grant no. 4052300, Harry F. LEE Vice-Chancellor's Discretionary Fund of The Chinese University of Hong Kong, grant no. 4930744, Steve H.L. YIM.

**Competing interests:** The authors have declared that no competing interests exist.

leading to unsustainable development [8]. Sustainable tourism development must consider environmental impacts in addition to economic benefits.

Air pollution has emerged as a major risk factor for travel, influencing tourist decisions and potentially reducing a country's tourism revenue. Unlike public health crises such as COVID-19, air pollution is a long-term chronic problem with more complex consequences for the tourism industry. For many years, air pollution has been linked to respiratory diseases, cardiovascular diseases, strokes, and lung cancer [9], raising concerns among tourists about air quality risks in tourist destinations. China has some of the worst $PM_{2.5}$ problems in the world [10]. Although smog pollution has long been a problem in China [11], public awareness of the problem did not rise significantly until 2013. Until this point, many people assumed they were encountering ordinary fog. However, China experienced its worst heavy smog pollution in half a century in 2013, resulting in frequent flight delays, park closures, and high respiratory infections [12]. As a result, the public's perception of air pollution has shifted, and this awareness has significantly influenced people's reactions to smog, potentially affecting domestic tourism.

It is critical to investigate the impact of air pollution on tourism development to achieve sustainable tourism development. Earlier research has shown that air pollution has both linear and non-linear effects on tourism development [13, 14]. These studies, however, lack a more specific and detailed examination, leaving the impact of air pollution on tourism industries at various stages of development largely unknown. We recognize that the factors affecting tourism are multifaceted, including aspects like the attractiveness of tourist destinations, transport accessibility, and so on. We propose that the impact of air pollution on tourism development may not be consistent but varies based on the various levels of tourism development. Specifically, in regions with well-established tourism sectors, the negative effects of air pollution may be more noticeable and manageable. On the other hand, in less-developed tourism areas, air pollution may not be the primary barrier to development. This perspective raises two primary research questions: during the initial stages of tourism development, does air pollution serve as the main obstacle? As the tourism sector progresses, how does the influence of air pollution shift?

To address these discrepancies, this study employs the Unconditional Quantile Regression (UQR) method to build a model that investigates the relationship between air pollution and tourism development. We can assess how environmental vulnerability affects the tourism industry symmetrically or asymmetrically by including tourism resources and transportation infrastructure as control variables and considering the interaction between air pollution and tourism resources. This approach addresses previous research limitations that ignore variable deviation, adding to the literature on the role of air quality in tourism development. The empirical method is the most important aspect of this study. To the best of our knowledge, the unconditional quantile regression approach has never been used in tourism research in China. This method lets us draw conclusions about our covariates, particularly our air pollution mitigation strategies. The findings from this study shed light on diverse perspectives of air pollution in cities with different levels of tourism development. By understanding the nuanced relationship between air pollution and varying stages of tourism development, the study aids in formulating more effective, stage-specific policies, thus contributing to the broader discourse on sustainable tourism.

## 2. Literature review

The quality of the air has a significant impact on travelers' destination choices. Klenosky [15] examined various push and pull factors and concluded that the atmosphere of a tourist destination is an essential factor in attracting visitors. Furthermore, environmental factors outperform other variables like tourist attractions and infrastructure [16, 17]. Environmental

psychology research has revealed that people's motivation to travel is heavily influenced by the satisfaction provided by the destination [18]. Simultaneously, people tend to avoid air pollution by taking avoidant actions [19, 20], making cities with better air quality more appealing as tourist destinations. In the context of China's tourism, studies have indicated that air quality affects both inbound and domestic tourism. However, international tourists exhibit greater concern about air pollution in China [13, 21]. The environmental Kuznets curve (EKC) theory posits that as a country's GDP per capita increases, so does the importance of the environment to its citizens [22]. The air pollution generated by China's rapid economic growth has heightened citizens' awareness of the need for clean air, which plays an increasingly significant role in determining travel preferences.

Numerous studies have examined the influence of air quality on tourism, yielding contrasting findings. Differences in research scope, estimation methods, and sample size selection have also resulted in discrepancies in findings among scholars. Some research has suggested no significant relationship between air pollution and tourist behavior. For instance, Sun et al. [23] conducted a fixed effect panel model and comparative analysis using data from 28 major cities in China, ultimately concluding that haze concentration had no notable impact on domestic travel. However, other studies have indicated that air pollution has a detrimental effect on tourism development. Tourists often evaluate their perception of emotional risk based on their opinion of a destination's air quality, and international tourists actively avoid areas with high levels of air pollution [21]. Beacon et al. [24] found that residents of the United States and Australia consider China's air pollution problem a significant travel risk, making pollution one of the primary deterrents for international tourists visiting the country. Dong et al. [25] analyzed data from 274 Chinese cities in 2009–2012 and uncovered that air pollution substantially reduces the number of international tourists to China. Nevertheless, the relationship between air pollution and tourist numbers may not be linear. Previous research has ignored the variations in air pollution's impact on tourism at different levels of development within the industry. Wang and Chen [13] employed the system generalized method of moments on data from 58 major tourism cities in China between 2004 and 2015, finding that air quality had an inverted U-shaped effect on tourist arrivals. Furthermore, Chien et al. [26] utilized the quantile autoregressive distributed lag (QARDL) technique to examine the non-linear association between air pollution and American tourism, both in the short and long term.

Although various studies have confirmed the dynamic relationship between air pollution and macroeconomic variables in the tourism industry, there is a paucity of related evidence in China. Early research in this area has primarily focused on traditional methods of investigating the correlation between crucial determinants of tourism. However, applying advanced techniques, such as UQR, remains a critical gap. UQR offers a more nuanced analysis by providing lower, middle, and upper quantile levels [27]. This method enables researchers to capture nonlinear, asymmetric relationships and structural breaks, which are impossible to estimate using traditional Ordinary Least Squares (OLS) techniques or the simple Quantile Regression (QR) method. By applying UQR, researchers can better understand the complexities of the relationship between air pollution and macroeconomic variables in China's tourism. This knowledge will be instrumental in informing more effective policies and interventions to mitigate the negative consequences of pollution on the industry.

## 3. Data and methodology

### 3.1. Data descriptions and model specification

**3.1.1. Dependent variables.** This study focuses on cities at the prefecture level in mainland China, using a panel data set spanning 284 cities from 2008 to 2018. In the context of

global tourism economic development, both tourist visits and tourism revenue have become increasingly important indicators for measuring a region's economic performance [28]. Furthermore, attracting tourist visits (TOV) and increasing tourism revenue (TOR) are among the primary goals pursued by local government decision-makers [29]. Hence, we choose tourist visits (TOV) and tourism revenue (TOR) as the dependent variables to assess the city's tourism development level.

**3.1.2. Independent variables.** $PM_{2.5}$ concentration is the core independent variable. There are primarily two options for obtaining China's $PM_{2.5}$ concentration data. One option is to obtain the data directly from the environmental department of China. However, this option has limitations, such as a short time span and limited spatial coverage. The other option, which is chosen in this study, involves estimating $PM_{2.5}$ concentration using nighttime light, LandScan, and other satellite images (the DMSP/OLS nighttime light remote sensing data and the landscape global population dynamic statistical analysis database). It provides $PM_{2.5}$ concentration data with extensive spatial coverage for a long period, allowing for the application of panel data analysis [30].

The surface $PM_{2.5}$ V5.GL.02 dataset, provided by van Donkelaar et al. [31], estimates ground-level fine particulate matter ($PM_{2.5}$) total and composite mass concentrations. The dataset combines NASA MODIS, MISR, and SeaWiFS aerosol optical depth (AOD) retrievals with the GEOS chemical transport model. It is then calibrated using Geographically Weighted Regression (GWR) to regional ground-based total and compositional mass observations. This study uses van Donkelaar et al.'s [31] dataset to calculate the mean and spatial variation of $PM_{2.5}$ concentrations in urban areas at various annual intervals.

Measuring air pollution at a suitable spatial scale is crucial to assessing local pollution accurately [32]. In China, there are four definitions for urban areas: municipal, urban area, urban district, and built-up areas. Cities are considered administrative units by the Chinese government, and they also include non-urban areas such as green spaces and the countryside. However, many studies ignore this issue and measure 'urban' air pollution in so-called cities, significantly affecting measurement accuracy. Although some studies employ the GHS build-up grids to define urban areas, the associated drawback is that panel data analysis for consecutive years becomes inapplicable because the GHS build-up grid dataset only provides data in the time slices of 2000 and 2014 [32, 33]. Therefore, this research chooses to use the artificial surface dataset provided by Gong et al. [34] to define the spatial coverage of urban areas and the sampling frame of $PM_{2.5}$. The artificial surface closely resembles the Chinese government's definition of urban built-up areas, which are defined as areas in urban administrative regions that have been developed and constructed, as well as basic municipal utilities and public facilities. This supports the use of built-up areas to assess urban air pollution.

**3.1.3. Control variables.** To better understand the relationship between air pollution and tourism development, we control for variables such as tourism resources, transportation infrastructure, and other socioeconomic data in this study. We employ the prefecture-level city's world cultural heritage, world natural heritage, 5A scenic spots, national scenic spots, national nature reserves, and national forest parks as proxies for the city's tourism resources (TOU). In line with China's 'Classification and Evaluation of Quality Rating of Tourist Attractions' [GB/T17775-2003] and previous studies [35, 36], we assign weights to these tourism resource types. Specifically, we allocate weights of 3, 3, 2, 1, 1, and 1 to world cultural heritage, world natural heritage, 5A scenic spots, national scenic spots, national nature reserves, and national forest parks, respectively.

The total length of roads (ROAD), airport grades (AIR), and railway station grades (RAIL) are all transportation infrastructure variables. We use the total length of roads in each city as a proxy for road transport. Airport grades are evaluated based on the Civil Aviation

Administration of China's classifications, which comprise five levels: 3C, 4C, 4D, 4E, and 4F. Moreover, we refer to the 'National Railway Station Grade Determination Method' to categorize the railway station grades into special, first, second, third, fourth, and fifth-level stations, aiding in estimating each station's transportation capacity. For cities with multiple facilities, such as different grades of airports or railway stations, we incorporate them into our model using an aggregated weighted score. Each facility is assigned a weight based on its grade, effectively capturing the overall transportation capacity of the city.

Previous research suggests that GDP, the ratio of the tertiary industry to GDP (TERIND), population size (POP), and the ratio of public fiscal expenditure to GDP (FISCAL) significantly impact urban air pollution [37–40]. Additionally, we have taken into account the effect of climatic variables such as temperature, wind speed, and precipitation on tourism [14]. Therefore, our model also controls for these variables to provide a more accurate estimation of the factors influencing tourism development in China.

All variables and data sources are presented in Table 1, and the descriptive statistics of the variables are presented in Table 2.

**Table 1. Data and their sources.**

| Variables | Definition | Data sources |
|---|---|---|
| Dependent variable (tourism) | | |
| TOV | Tourist visits | National Economic and Social Development Statistical Bulletin, EPS Global Statistical Data Analysis Platform, Chinese City Statistical Yearbooks, Chinese City Tourism Bureau official websites |
| TOR | Tourism revenue | |
| Independent variable (air pollution) | | |
| $PM_{2.5}$ | $PM_{2.5}$ concentrations | Van Donkelaar et al. (2021) |
| Control variables | | |
| TOU | Tourism resources, indicated by the weighted sum of world cultural heritage sites, world natural heritage sites, 5A scenic spots, national scenic spots, national nature reserves, and national forest parks | National Economic and Social Development Statistical Bulletin, EPS Global Statistical Data Analysis Platform, Chinese City Statistical Yearbooks, Chinese City Tourism Bureau official websites, and the National Tibetan Plateau Data Center (TPDC) |
| ROAD | Road transport, indicated by the total length of the roads | |
| AIR | Air transport, indicated by the airport grades | |
| RAIL | Rail transport, indicated by the train station grades | |
| GDP | GDP, indicated by the total value of goods and services produced in an area | |
| TERIND | Tertiary industry, indicated by the ratio of the tertiary industry to GDP | |
| POP | Population size | |
| FISCAL | Fiscal expenditure, indicated by the ratio of public fiscal expenditure to GDP | |
| TEMP | Temperature, indicated by the instantaneous near surface (2 m) air temperature (°C) | |
| WIND | Wind, indicated by the instantaneous near surface (10 m) wind speed (m s$^{-1}$) | |
| PREC | Precipitation, indicated by the 3-hourly mean (from -3.0 hr to 0.0 hr) precipitation rate (mm hr$^{-1}$) | |

**Table 2. Descriptive statistics of the variables employed in this study.**

| Variable | Obs | Mean | Std. dev. | Min | Max |
|---|---|---|---|---|---|
| $\ln PM_{2.5}$ | 3124 | 3.888 | 0.319 | 2.784 | 4.836 |
| $\ln TOV$ | 3124 | 7.410 | 1.079 | 1.826 | 10.998 |
| $\ln TOR$ | 3124 | 4.962 | 1.284 | 0.359 | 9.904 |
| $\ln TOU$ | 3124 | 12.613 | 11.580 | 0 | 128 |
| $\ln ROAD$ | 3124 | 9.233 | 0.694 | 6.293 | 11.967 |
| $AIR$ | 3124 | 1.985 | 3.298 | 0 | 18 |
| $RAIL$ | 3124 | 15.230 | 14.931 | 0 | 119 |
| $\ln GDP$ | 3124 | 15.476 | 1.199 | 12.321 | 19.605 |
| $TERIND$ | 3124 | 44.562 | 11.461 | 8.580 | 92.010 |
| $\ln POP$ | 3124 | 4.637 | 0.781 | 2.715 | 7.810 |
| $FISCAL$ | 3124 | 0.171 | 0.110 | 0.010 | 2.702 |
| $TEMP$ | 3124 | 14.49 | 5.32 | -2.13 | 25.55 |
| $WIND$ | 3124 | 2.32 | 0.75 | 0.47 | 6.27 |
| $PREC$ | 3124 | 0.12 | 0.07 | 0.008 | 0.38 |

## 3.2. Model specification

The basic model specification of this study is as follows:

$$tourism_{it} = \alpha_{it} + \beta PM_{2.5it} + \beta X_{it} + + \eta_t + \mu_i + \varepsilon_{it} \tag{1}$$

Where $i$ represents the city and $t$ denotes the year. $tourism_{it}$ is the tourism development variable, including tourist visits (TOV) and tourism revenue (TOR). $\beta$ is the coefficient. $PM_{2.5it}$ is the air pollution variable. $X_{it}$ is a set of control variables related to tourism development. $\mu_i$ is the fixed effect of the city, $\eta_t$ is time-fixed effects of the year. $\varepsilon_{it}$ is the error term.

The QR method is used in this study. The traditional OLS method assumes that the covariate of interest has a sustained effect on different levels of the outcome variable, indicating only the covariate's impact on the average value of the outcome variable without taking into account the heterogeneity of unobserved city effects [41]. This study considers the QR method, which allows for examining how the effect of quantiles of the outcome distribution may differ. This approach is consistent with our research into whether air pollution affects tourism differently in different cities, particularly between low and high levels of tourism development.

Furthermore, the traditional alternative method for dealing with the outcome variable's heterogeneity is to divide the sample and then compare and analyze the split sub-samples. However, as Alan et al. [42] argue, this approach causes 'sample truncation' issues and may yield invalid results due to sample selection bias. QR, on the other hand, uses the entire sample and avoids such issues. Furthermore, our results resist estimation biases because QR does not rely on distribution assumptions.

It is important to note that the Conditional Quantile Regression (CQR) method proposed by Koenker and Bassett [41] may result in biased estimates based on too many, or even unnecessary, individual characteristics. UQR, as an extension and supplement to CQR, provides a general marginal effect of explanatory variables on the explained variable. Moreover, it supports controlling fixed effects in the model. Therefore, we revise our Model 1 to the following UQR model:

$$Q_\tau(tourism_{it}) = \alpha_\tau + \beta_\tau PM_{2.5it} + X_{it} + \eta_t + \mu_i + \varepsilon_{it} \tag{2}$$

Where $Q_\tau$ *(tourism$_{it}$)* is the τ-th quantile of the distribution of the tourism variable (including tourist visits and tourist revenue). $\beta_\tau$ is our main estimation parameter, representing the impact of each explanatory variable on the τ quantiles.

## 4. Statistical results and analyses

### 4.1. Unit root test

For our panel unit root analysis, we have opted for the Im-Pesaran-Shin (IPS) test and the Harris-Tzavalis (HT) test. The IPS test prescribes certain minimums for the time dimension (T). Specifically, for models with only panel-specific means, a balanced dataset must have at least five time periods; this increases to six if time trends are also present. Should a dataset fail to meet these minimums, the p-value for Z_t-tilde-bar will not be computed. As our dataset's time dimensionality comfortably meets these criteria, we primarily rely on the IPS test. The HT test also holds relevance in our analysis because it operates under the premise of a fixed number of time periods, T, in direct contrast to tests based on the assumption of T without bound.

We have also considered other unit root tests. While prevalent, the Levin-Lin-Chu (LLC) test is best suited for datasets of moderate size, with the standard being at least 25 observations per individual. Our dataset, with a greater number of individuals but fewer observations per individual than this standard, is incompatible with the LLC test. Additionally, we find the Fisher-type tests and the Hadri test, which assume that T extends to infinity, to be ill-suited for our data. The same applies to Breitung's test, which requires that T, followed by N, increase without limit to achieve an asymptotically normal distribution.

Table 3 shows that most variables have passed the unit root test. However, ln*GDP* and ln*TER* do not pass the Harris-Tzavalis (HT) test. Consequently, we perform the first

**Table 3. Unit root test results.**

| Variables | Im-Pesaran-Shin (IPS) test | Harris-Tzavalis (HT) test |
|---|---|---|
| | Z-t-tilde-bar | rho |
| ln*TOV* | -6.7208*** | -0.1502*** |
| ln*TOR* | -6.9642*** | -0.0922*** |
| ln*PM$_{2.5}$* | -2.1604*** | 0.3781** |
| ln*TOU* | — | 0.2622*** |
| ln*ROAD* | -8.784*** | -0.3128*** |
| *AIR* | — | 0.3515*** |
| *RAIL* | — | 0.3158*** |
| ln*GDP* | 0.8785 | 0.415 |
| D_ln*GDP* | -17.9747*** | -0.0625*** |
| *TERIND* | -4.1426*** | 0.5247 |
| D_*TERIND* | -19.3407*** | 0.1249*** |
| ln*POP* | -5.6845*** | 0.3013*** |
| *FISCAL* | -10.5376*** | -0.0493*** |
| *TEMP* | -18.05*** | 0.0594*** |
| *WIND* | -11.5098*** | 0.2847*** |
| *PREC* | -23.0724*** | -0.2491*** |

*** p < 0.01

** p < 0.05.

differencing on these variables. The differenced variables, denoted as D_ln$GDP$ and D_ln$TER$, pass the unit root test and are used in the subsequent regression analysis.

## 4.2. Non-linear effect of air pollution on tourism

We conduct UQR to examine the relationship between air pollution (PM$_{2.5}$) and tourist visits (TOV). Table 4 presents the UQR estimation results, with columns (1)–(5) reporting the conditional concentration distribution at the 10th (Q10), 25th (Q25), 50th (Q50), 75th (Q75), and 90th (Q90) percentiles, and column (6) summarizing the ordinary least squares (OLS) regression estimates. Those percentiles help reveal the effects of air pollution on tourism across different levels of tour visits. We categorize those levels as low ($\tau \leq 0.25$), medium ($0.25 < \tau \leq 0.75$), and high ($\tau > 0.75$).

**Table 4. UQR estimation results for PM$_{2.5}$ on tourist visits (TOV).**

| Variables | (1) Q10 | (2) Q25 | (3) Q50 | (4) Q75 | (5) Q90 | (6) FE |
|---|---|---|---|---|---|---|
| ln$PM_{2.5}$ | 1.440*** | -0.098 | -0.951*** | -0.464* | 0.172 | -0.167** |
| | (0.433) | (0.259) | (0.217) | (0.245) | (0.239) | (0.068) |
| ln$TOU$ | -0.058*** | -0.046*** | -0.012 | 0.020* | 0.044*** | -0.001 |
| | (0.012) | (0.008) | (0.008) | (0.012) | (0.013) | (0.003) |
| ln$ROAD$ | 0.750* | 0.315 | -0.244 | -0.562*** | -0.518*** | -0.009 |
| | (0.420) | (0.210) | (0.197) | (0.162) | (0.174) | (0.078) |
| $AIR$ | 0.065 | -0.099*** | -0.029 | 0.110** | 0.146 | 0.008 |
| | (0.111) | (0.025) | (0.027) | (0.056) | (0.093) | (0.013) |
| $RAIL$ | 0.059* | -0.038** | -0.053*** | -0.026 | 0.014 | -0.009** |
| | (0.035) | (0.016) | (0.011) | (0.018) | (0.025) | (0.004) |
| D_ln$GDP$ | 0.082 | -0.130 | 0.007 | 0.171* | -0.090 | 0.023 |
| | (0.238) | (0.122) | (0.095) | (0.102) | (0.162) | (0.033) |
| D_$TERIND$ | 0.022* | 0.005 | -0.000 | 0.000 | -0.006 | 0.002 |
| | (0.013) | (0.007) | (0.005) | (0.006) | (0.006) | (0.002) |
| ln$POP$ | -0.105 | 0.019 | -0.040 | 0.095 | 0.675** | -0.033 |
| | (0.352) | (0.183) | (0.140) | (0.193) | (0.289) | (0.054) |
| $FISCAL$ | 0.198 | -0.084 | -0.133 | -0.155 | -0.237 | -0.099 |
| | (0.542) | (0.297) | (0.238) | (0.338) | (0.186) | (0.085) |
| $TEMP$ | -0.096 | -0.038 | 0.003 | -0.104** | 0.030 | -0.025 |
| | (0.090) | (0.066) | (0.052) | (0.052) | (0.054) | (0.017) |
| $WIND$ | -0.051 | -0.069 | 0.078 | 0.017 | -0.079 | 0.030 |
| | (0.151) | (0.109) | (0.082) | (0.095) | (0.084) | (0.027) |
| $PREC$ | -0.180 | 1.565** | 0.765 | -0.388 | -1.271 | 0.084 |
| | (1.129) | (0.720) | (0.740) | (0.755) | (0.811) | (0.189) |
| Constant | -4.682 | 6.183** | 14.419*** | 16.208*** | 8.618*** | 8.012*** |
| | (4.872) | (2.565) | (2.273) | (2.171) | (2.393) | (0.857) |
| Observations | 2,840 | 2,840 | 2,840 | 2,840 | 2,840 | 2,840 |
| R-squared | 0.635 | 0.663 | 0.683 | 0.656 | 0.611 | 0.822 |
| No. of cities | 284 | 284 | 284 | 284 | 284 | 284 |

Note: TOV is the dependent variable.

*** $p < 0.01$

** $p < 0.05$

* $p < 0.1$.

The UQR results show significant variations in the impacts of air pollution on tourism across the different tourism development levels. In cities with low TOV (Q10), higher $PM_{2.5}$ is associated with more TOV. Holding other factors constant, a 1% increase in $PM_{2.5}$ stimulates a 1.44% growth in TOV. Notably, the association is stronger for areas with fewer TOV, as reflected by the higher estimated coefficients of $PM_{2.5}$ toward Q10. This finding contradicts Shah et al. [43]. Nevertheless, this result is primarily observed in the early phases of tourism development. At higher quantiles of the regression analysis, a negative correlation between $PM_{2.5}$ and TOV becomes significant. Our analysis suggests an alternative narrative: during the initial phases of tourism development, the expansion of infrastructure, which may elevate air pollution levels, might simultaneously enhance tourism by improving the accessibility of the area [44]. This relationship implies that the early stages of tourism development may be accompanied by increased pollution and tourism growth.

The UQR results further show that the positive correlation between $PM_{2.5}$ and TOV at Q10 turns negative around Q50 and becomes weaker after Q75; it is insignificant at Q90. To comprehend these complex changes in the quantile coefficients, we present the UQR estimated coefficients of $PM_{2.5}$ on TOV for each quantile (Fig 1). Fig 1 shows a trend of moving toward negative, then back to positive and finally dropping to below zero. When TOV reaches Q25, higher $PM_{2.5}$ reduces TOV. This finding concurs with Deng et al. [45], Wang et al. [46], and Wang and Chen [13], who suggest that high $PM_{2.5}$ concentrations reduce tourist safety and deter tourist visits once air pollution has reached a certain level. Additionally, our findings show that the obstructing effect of air pollution on tourism is less pronounced in cities with higher TOV. Such a phenomenon can be explained by the fact that areas with better tourism development possess more well-established tourism facilities. The industrial structure there has also undergone optimization and upgrading [47], which can mitigate the negative impact

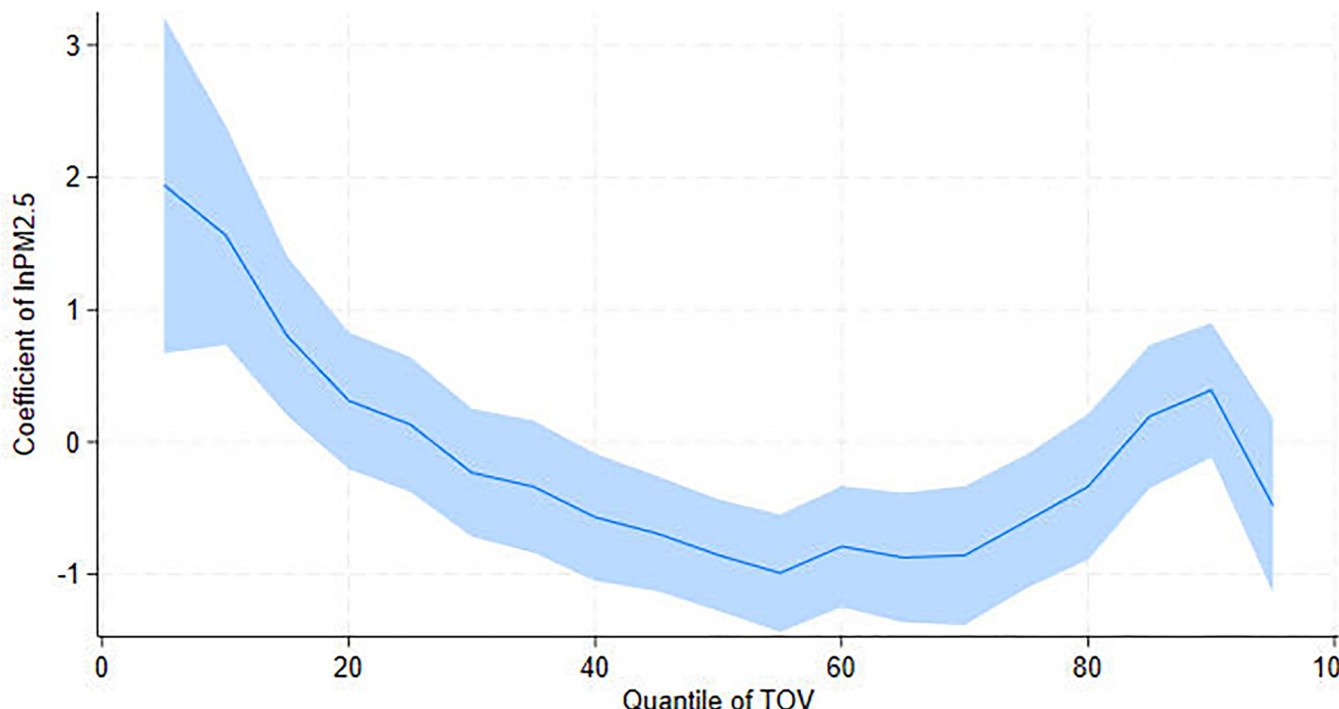

**Fig 1. UQR coefficient distribution for the effect of $PM_{2.5}$ on tourist visits (TOV).**

of air pollution on tourism. This finding highlights the importance of the specific context and tourism development levels in examining the effects of air pollution on tourism.

Fig 2 reveals that tourism resources (TOU) significantly influence tourism. Specifically, TOU is negatively correlated with TOV at low percentiles but positively correlated with TOV at high percentiles. This implies that tourism resources alone may not be sufficient to attract more tourists to areas with low TOV. However, when TOV reaches the 63% percentile, TOU positively affects TOV. The impact of tourism resources on tourism remains unclear in the literature. Proponents of the tourism-led economic growth hypothesis argue that according to factor endowment theory, regions with abundant tourism resources possess greater comparative advantages [48, 49]. They can attract more tourists and foster tourism. This perspective highlights the promotion effect of tourism resources on tourist visits in places at medium and high tourism development levels. Supporters of the resource curse hypothesis, on the other hand, argue that overreliance on tourism resources can lead to the Dutch disease effect (i.e., resource prosperity leading to deindustrialization) [50], resulting in unfavorable economic development [51–53]. Additionally, economies that rely heavily on natural resources, such as specialized tourist cities, are more vulnerable to external shocks [54]. This vulnerability may explain why tourism resources obstruct areas with low TOV.

Several interesting patterns have emerged from our analysis of the impact of transportation infrastructure on TOV. ROAD shows a slight positive influence on TOV at Q10, yet exhibits negative impacts at higher percentiles, Q75 and Q90. Conversely, AIR demonstrates a negative impact on TOV at Q25 but shifts to a positive influence at Q75. Meanwhile, RAIL positively affects TOV at Q10, but has a negative effect at Q25 and Q50, and shows no significant impact at Q75 and Q90.

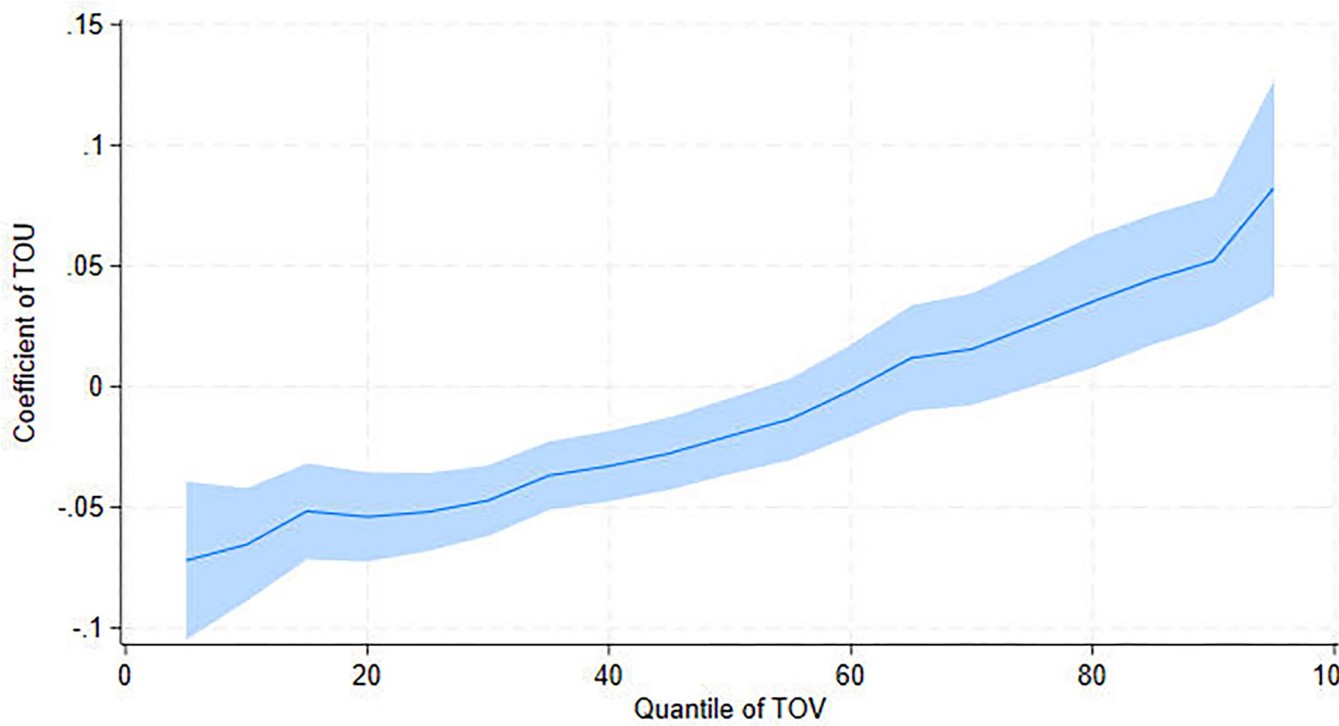

**Fig 2. UQR coefficient distribution for the effect of tourism resource (TOU) on tourist visits (TOV).**

Several relationships stand out when examining other control variables. POP demonstrates a positive effect on TOV in high TOV areas (Q90). This effect likely stems from the agglomeration phenomenon prevalent in tourist hotspots, which facilitates the expansion of tourism-related services and infrastructure. TERIND exerts a modest positive influence on TOV at Q10, suggesting its supportive role in the initial stages of tourism development. TEMP negatively impacts TOV at Q75. PREC, on the other hand, shows a positive effect on TOV at Q25. Regarding FISCAL, lnGDP, and WIND, these variables do not exert a significant influence on TOV.

To investigate the impact of air pollution on tourism further, we use UQR to examine the effect of $PM_{2.5}$ on another indicator of tourism development–tourism revenue (TOR). Columns (1)—(5) in Table 5 report the conditional concentration distributions of 10th (Q10), 25th (Q25), 50th (Q50), 75th (Q75), and 90th (Q90) percentiles, while column (6) summarizes the OLS regression estimates. The findings are consistent with the UQR estimate coefficients of $PM_{2.5}$ on TOV, confirming the link between air pollution and tourism. Notably, $PM_{2.5}$ has a positive effect on TOR at Q10 but has a significant negative effect at Q50 (Fig 3). At Q75 and Q90, however, no effect is observed. This implies that air pollution can significantly reduce tourism revenue in areas with medium TOR but has no effect in areas with high TOR. The UQR estimate coefficients of TOU on TOR are shown in Fig 4, which mirrors the effects of TOU on TOV (Fig 2). Briefly, TOU has a suppressive effect on TOR at low percentiles. TOU, on the other hand, has a promoting effect on TOR at high percentiles.

## 4.3. Interaction effect of air pollution and tourism resources on tourism development

Our statistical findings emphasize the importance of tourism resources in tourism development. We investigate the interaction effect of $PM_{2.5}$ and TOU on tourism development to delve deeper into this relationship. Table 6 shows the interaction effect on TOV in various percentiles. The interaction effect negatively impacts TOV at Q10 and positively on TOV at Q50 and Q75. However, at high percentiles, the interaction effect becomes insignificant. These findings imply that tourism resources can reduce the positive correlation between air pollution and tourist visits in low TOV areas. In contrast, tourism resources can help mitigate the negative effects of air pollution on tourist visits in the middle to upper percentiles. Fig 5 depicts the non-linear interaction effect of $PM_{2.5}$ and TOU on TOV. This implies that only when tourism development reaches a certain level will tourism resources be able to partially offset the negative impact of air pollution on tourist visits. Such an 'offsetting effect' is the greatest at the 58th percentile.

Table 7 shows the interaction effects of $PM_{2.5}$ and TOU on TOR, similar to the ones for TOV (Table 6). At Q10, the interaction effects are negatively significant, but at Q50, they are positively significant. Fig 6 depicts the inverted U-shaped curve of the interaction effects of $PM_{2.5}$ and TOU on TOR. The curve supports the idea that once tourism development reaches a certain level, tourism resources can mitigate the obstructive effect of air pollution on tourism.

## 4.4. Linear trends and non-linear effects

To evaluate potential linear trends and non-linear effects within our model, we apply median regression at the 50th percentile. As a specific form of quantile regression, median regression is adept at circumventing issues related to outliers in data, a common challenge with mean regression models that utilize fixed effects. Moreover, this method does not necessitate the strict adherence of data to a normal distribution, offering a robust alternative for analyzing

**Table 5. UQR estimation results for PM$_{2.5}$ on tourism revenue (TOR).**

| Variables | (1) Q10 | (2) Q25 | (3) Q50 | (4) Q75 | (5) Q90 | (6) FE |
|---|---|---|---|---|---|---|
| ln$PM_{2.5}$ | 1.559*** | -0.281 | -1.595*** | 0.119 | 0.265 | -0.233*** |
| | (0.520) | (0.342) | (0.248) | (0.328) | (0.312) | (0.073) |
| ln$TOU$ | -0.070*** | -0.049*** | -0.016 | 0.022* | 0.056*** | -0.003 |
| | (0.014) | (0.010) | (0.010) | (0.013) | (0.016) | (0.003) |
| ln$ROAD$ | 0.848** | 0.416 | -0.338* | -0.538*** | -0.650*** | 0.045 |
| | (0.362) | (0.303) | (0.201) | (0.189) | (0.167) | (0.067) |
| $AIR$ | -0.014 | -0.039 | -0.054 | 0.137** | 0.222* | 0.011 |
| | (0.102) | (0.082) | (0.036) | (0.065) | (0.128) | (0.013) |
| $RAIL$ | 0.054 | -0.020 | -0.057*** | -0.034 | 0.023 | -0.007 |
| | (0.042) | (0.026) | (0.014) | (0.021) | (0.033) | (0.005) |
| D_ln$GDP$ | 0.034 | 0.231 | -0.045 | 0.022 | -0.266 | -0.010 |
| | (0.229) | (0.142) | (0.109) | (0.124) | (0.186) | (0.032) |
| D_$TERIND$ | 0.023 | 0.011 | -0.001 | -0.008 | -0.006 | 0.005** |
| | (0.015) | (0.008) | (0.006) | (0.007) | (0.008) | (0.002) |
| ln$POP$ | -1.206*** | -0.066 | 0.242 | 0.511** | 0.812** | 0.037 |
| | (0.248) | (0.268) | (0.162) | (0.248) | (0.354) | (0.056) |
| $FISCAL$ | 0.318 | 0.276 | -0.092 | 0.006 | -0.208 | -0.081 |
| | (0.595) | (0.370) | (0.225) | (0.371) | (0.232) | (0.093) |
| $TEMP$ | -0.052 | -0.093 | -0.062 | -0.045 | 0.079 | -0.072*** |
| | (0.115) | (0.085) | (0.058) | (0.064) | (0.087) | (0.021) |
| $WIND$ | 0.155 | -0.020 | -0.014 | 0.067 | -0.129 | 0.017 |
| | (0.178) | (0.128) | (0.096) | (0.110) | (0.119) | (0.027) |
| $PREC$ | 0.380 | 0.486 | 0.662 | -0.470 | -0.056 | -0.173 |
| | (1.215) | (0.974) | (0.805) | (0.926) | (1.072) | (0.219) |
| Constant | -4.439 | 4.099 | 15.368*** | 8.553*** | 5.518* | 5.552*** |
| | (4.306) | (3.597) | (2.344) | (2.633) | (2.822) | (0.800) |
| Observations | 2,840 | 2,840 | 2,840 | 2,840 | 2,840 | 2,840 |
| R-squared | 0.590 | 0.651 | 0.688 | 0.661 | 0.642 | 0.856 |
| No. of cities | 284 | 284 | 284 | 284 | 284 | 284 |

Note: TOR is the dependent variable.

*** $p < 0.01$

** $p < 0.05$

* $p < 0.1$.

diverse datasets [41]. This technique effectively reduces the disproportionate influence of cities exhibiting extremely high or low levels of TOV or TOR on the aggregated findings.

Table 8 presents an analysis of the impact of air pollution on TOV and TOR, incorporating both quadratic terms and linear trends for a comprehensive overview. Columns 1 and 2 feature the baseline median regression outcomes, while columns 3 and 4 extend these findings by including quadratic terms. Columns 5 and 6 explore the presence of linear trends. The analysis reveals the absence of quadratic air pollution interactions concerning TOV and TOR, suggesting a lack of significant non-linear effects. However, the linear trend analysis indicates a notable positive linear trend in the impact of air pollution on TOV. This trend implies that although air pollution initially detracts from tourist visits, its adverse impact gradually diminishes over time. In contrast, no similar linear trend emerges for TOR, indicating that while air

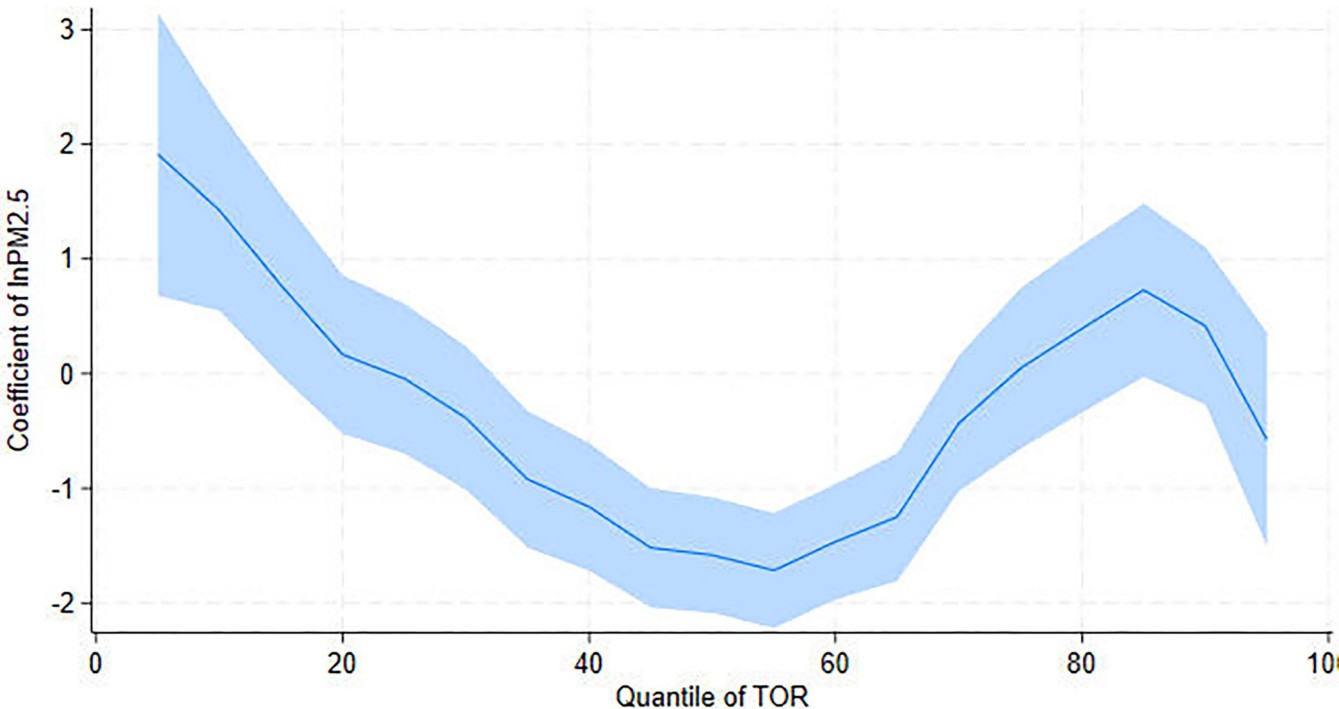

**Fig 3. UQR coefficient distribution for the effect of PM$_{2.5}$ on tourism revenue (TOR).**

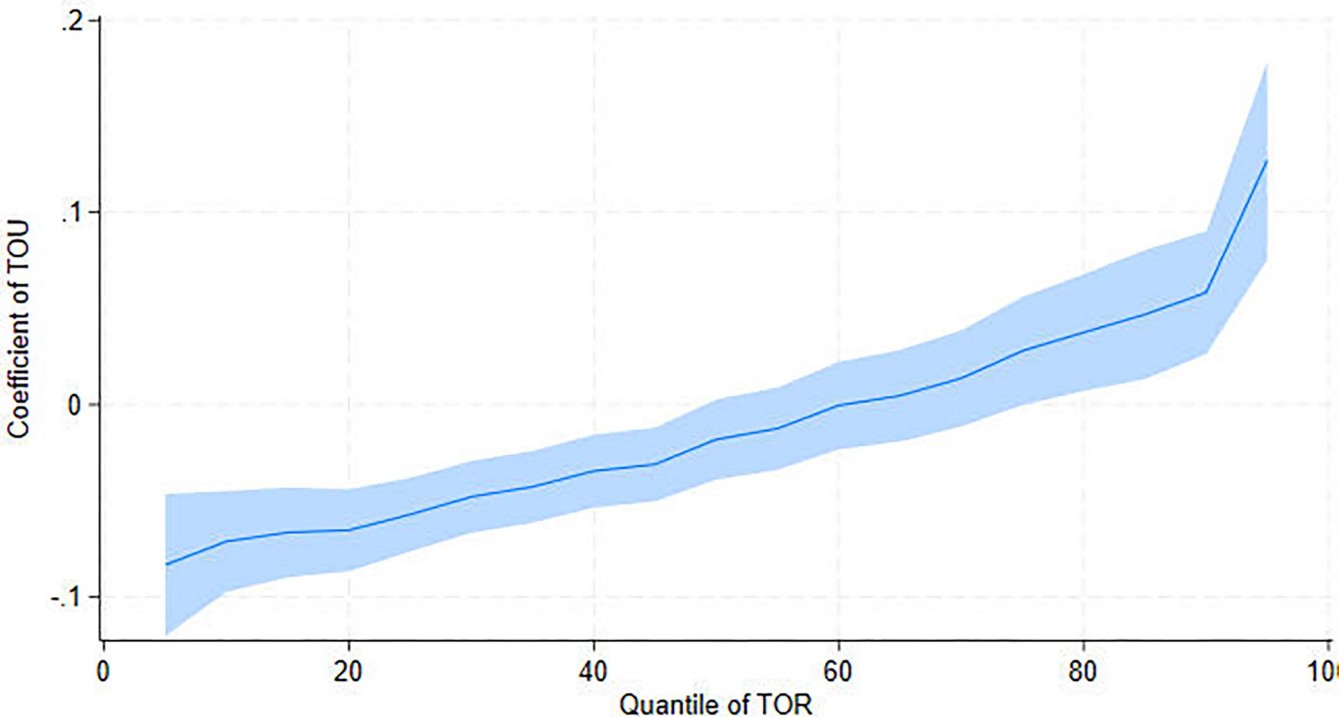

**Fig 4. UQR coefficient distribution for the effect of tourism resource (TOU) on tourism revenue (TOR).**

**Table 6. UQR estimation results for the interaction effects between $PM_{2.5}$ and tourism resources (TOU) on tourist visits (TOV).**

| Variables | (1) Q10 | (2) Q25 | (3) Q50 | (4) Q75 | (5) Q90 | (6) FE |
|---|---|---|---|---|---|---|
| ln$PM_{2.5}$ | 1.903*** | -0.089 | -1.587*** | -1.048*** | 0.393 | -0.219** |
|  | (0.526) | (0.318) | (0.256) | (0.289) | (0.341) | (0.087) |
| ln$TOU$ | 0.060 | -0.043 | -0.174*** | -0.128*** | 0.100 | -0.015 |
|  | (0.043) | (0.035) | (0.032) | (0.041) | (0.082) | (0.014) |
| ln$PM_{2.5}$xln$TOU$ | -0.032*** | -0.001 | 0.044*** | 0.041*** | -0.015 | 0.004 |
|  | (0.011) | (0.009) | (0.009) | (0.011) | (0.021) | (0.004) |
| ln$ROAD$ | 0.720* | 0.315 | -0.202 | -0.523*** | -0.533*** | -0.005 |
|  | (0.415) | (0.210) | (0.191) | (0.159) | (0.174) | (0.078) |
| $AIR$ | 0.057 | -0.099*** | -0.019 | 0.119** | 0.142 | 0.009 |
|  | (0.111) | (0.025) | (0.027) | (0.055) | (0.093) | (0.013) |
| RAIL | 0.060* | -0.038** | -0.054*** | -0.027 | 0.014 | -0.009** |
|  | (0.035) | (0.016) | (0.011) | (0.017) | (0.025) | (0.004) |
| D_ln$GDP$ | 0.103 | -0.130 | -0.022 | 0.145 | -0.080 | 0.020 |
|  | (0.238) | (0.122) | (0.093) | (0.099) | (0.162) | (0.033) |
| D_$TERIND$ | 0.022* | 0.005 | -0.000 | 0.000 | -0.006 | 0.002 |
|  | (0.013) | (0.007) | (0.005) | (0.006) | (0.006) | (0.002) |
| ln$POP$ | -0.118 | 0.018 | -0.022 | 0.112 | 0.669** | -0.032 |
|  | (0.353) | (0.184) | (0.137) | (0.200) | (0.286) | (0.054) |
| $FISCAL$ | 0.172 | -0.084 | -0.098 | -0.123 | -0.249 | -0.096 |
|  | (0.540) | (0.297) | (0.235) | (0.332) | (0.186) | (0.084) |
| $TEMP$ | -0.092 | -0.038 | -0.003 | -0.109** | 0.032 | -0.026 |
|  | (0.090) | (0.066) | (0.052) | (0.052) | (0.054) | (0.017) |
| $WIND$ | -0.034 | -0.069 | 0.056 | -0.003 | -0.071 | 0.028 |
|  | (0.152) | (0.109) | (0.081) | (0.095) | (0.084) | (0.027) |
| $PREC$ | -0.209 | 1.565** | 0.805 | -0.351 | -1.284 | 0.087 |
|  | (1.132) | (0.720) | (0.730) | (0.752) | (0.811) | (0.189) |
| Constant | -6.131 | 6.158** | 16.407*** | 18.034*** | 7.927*** | 8.179*** |
|  | (4.906) | (2.616) | (2.243) | (2.241) | (2.629) | (0.869) |
| Observations | 2,840 | 2,840 | 2,840 | 2,840 | 2,840 | 2,840 |
| R-squared | 0.636 | 0.663 | 0.687 | 0.660 | 0.612 | 0.823 |
| No. of cities | 284 | 284 | 284 | 284 | 284 | 284 |

Note: TOV is the dependent variable.

*** $p < 0.01$

** $p < 0.05$

* $p < 0.1$.

pollution might initially dampen the public's travel enthusiasm, this concern diminishes over time. Despite this, such a trend does not translate into a corresponding increase in tourism revenue, highlighting that an increase in willingness to visit more polluted cities does not necessarily result in higher economic benefits from tourism.

## 5. Discussions and policy implications

This study demonstrates the non-linear relationship between air pollution and tourism, revealing that the relationship depends on the tourism development level. Contrary to popular belief, air pollution does not consistently harm the tourism industry. Based on our statistical findings, we provide policy recommendations for enhancing tourism development as follows:

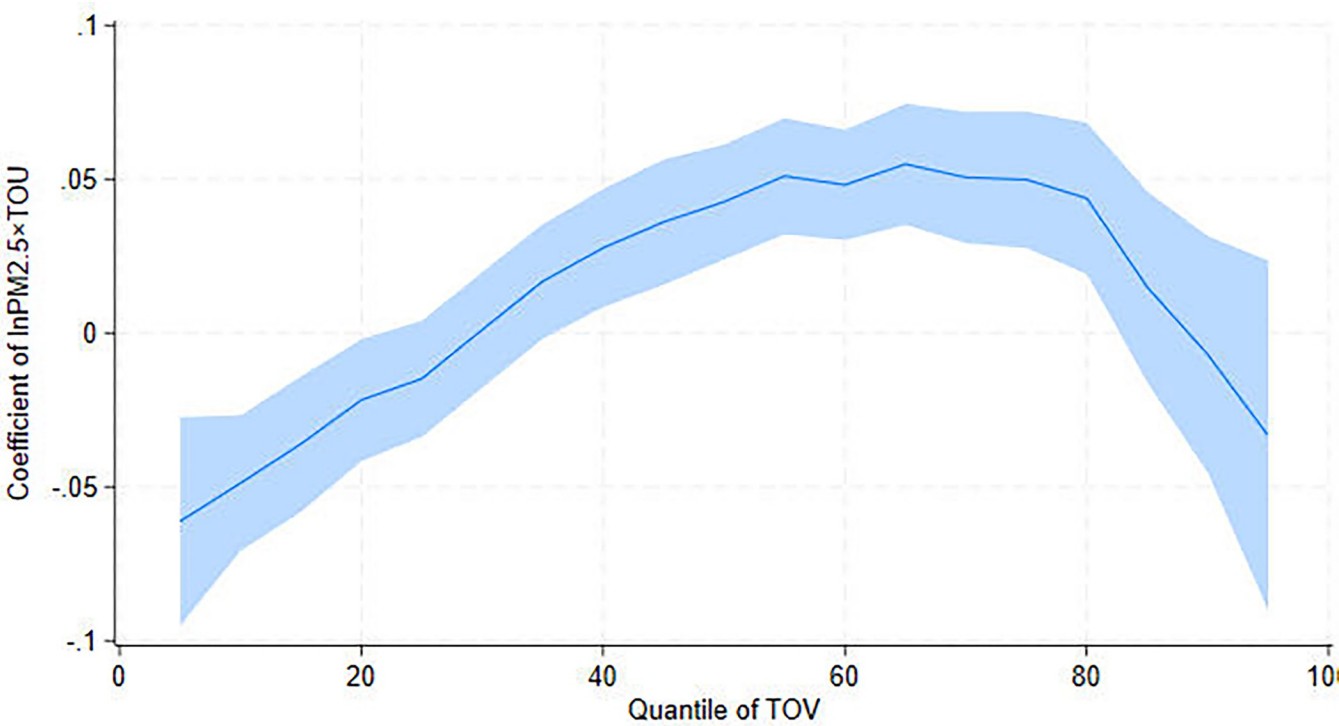

**Fig 5. UQR coefficient distribution for the interaction effects between PM$_{2.5}$ and tourism resource (TOU) on tourist visits (TOV).**

### 5.1. Strategic development in tourism to cope with air pollution

According to the resource curse hypothesis, tourism resources can have a negative impact on tourism in cities with low tourism development levels [55]. Tourism resources, on the other hand, can offset the negative effects of air pollution on tourism, indicating a complex relationship. It is not ideal to rely on road and railway transport for tourism development, while air transport might be prioritized. Transportation infrastructure development can promote tourism while contributing to air pollution and stifling tourism growth. As a result, cities that are growing their tourism should prioritize air quality. Tourism development varies between developed and developing Chinese cities as a pillar industry. Traditional, extensive tourism development is undesirable because excessive energy consumption and resource development can lead to economic decline due to tourism's environmental externalities [29, 56].

### 5.2. Developing sustainable tourism through comprehensive transportation systems

Transportation infrastructure is critical for the growth of tourism, but balancing its construction with long-term economic and environmental benefits is also vital. When tourism development lags, policymakers should prioritize tourism product enhancement over infrastructure construction [57]. Excessive infrastructure can result in waste and pollution of the environment. Tourism can help other industries by establishing regional aviation transportation networks when it grows. Cities can develop comprehensive all-domain transportation systems to promote tourism transformation and high-quality development. This approach integrates various modes of transportation, such as road, rail, air, and water, to provide tourists with seamless connectivity while minimizing transportation's environmental externalities.

**Table 7. UQR estimation results for the interaction effects between PM$_{2.5}$ and tourism resources (TOU) on tourist revenue (TOR).**

| Variables | (1) Q10 | (2) Q25 | (3) Q50 | (4) Q75 | (5) Q90 | (6) FE |
|---|---|---|---|---|---|---|
| ln$PM_{2.5}$ | 2.099*** | -0.531 | -2.313*** | -0.220 | 0.836* | -0.245*** |
|  | (0.617) | (0.416) | (0.288) | (0.387) | (0.447) | (0.093) |
| ln$TOU$ | 0.067 | -0.112** | -0.198*** | -0.065 | 0.201* | -0.006 |
|  | (0.049) | (0.048) | (0.037) | (0.055) | (0.108) | (0.014) |
| ln$PM_{2.5}$xln$TOU$ | -0.038*** | 0.017 | 0.050*** | 0.024 | -0.040 | 0.001 |
|  | (0.013) | (0.013) | (0.010) | (0.014) | (0.028) | (0.003) |
| ln$ROAD$ | 0.813** | 0.432 | -0.290 | -0.516*** | -0.688*** | 0.046 |
|  | (0.360) | (0.302) | (0.195) | (0.188) | (0.167) | (0.067) |
| $AIR$ | -0.022 | -0.035 | -0.042 | 0.142** | 0.212 | 0.011 |
|  | (0.101) | (0.082) | (0.036) | (0.065) | (0.129) | (0.013) |
| $RAIL$ | 0.055 | -0.020 | -0.058*** | -0.034* | 0.024 | -0.007 |
|  | (0.041) | (0.026) | (0.014) | (0.020) | (0.033) | (0.005) |
| D_ln$GDP$ | 0.058 | 0.220 | -0.077 | 0.007 | -0.240 | -0.010 |
|  | (0.228) | (0.140) | (0.108) | (0.125) | (0.182) | (0.033) |
| D_$TERIND$ | 0.023 | 0.011 | -0.001 | -0.008 | -0.006 | 0.005** |
|  | (0.015) | (0.008) | (0.006) | (0.007) | (0.008) | (0.002) |
| ln$POP$ | -1.221*** | -0.058 | 0.263 | 0.521** | 0.796** | 0.038 |
|  | (0.250) | (0.266) | (0.164) | (0.253) | (0.345) | (0.056) |
| $FISCAL$ | 0.288 | 0.290 | -0.052 | 0.025 | -0.240 | -0.081 |
|  | (0.591) | (0.371) | (0.221) | (0.368) | (0.232) | (0.093) |
| $TEMP$ | -0.046 | -0.096 | -0.069 | -0.049 | 0.085 | -0.073*** |
|  | (0.115) | (0.085) | (0.057) | (0.064) | (0.087) | (0.021) |
| $WIND$ | 0.174 | -0.029 | -0.039 | 0.055 | -0.109 | 0.016 |
|  | (0.178) | (0.128) | (0.097) | (0.111) | (0.119) | (0.027) |
| $PREC$ | 0.346 | 0.502 | 0.706 | -0.449 | -0.091 | -0.172 |
|  | (1.212) | (0.973) | (0.805) | (0.924) | (1.068) | (0.218) |
| Constant | -6.127 | 4.882 | 17.614*** | 9.611*** | 3.733 | 5.591*** |
|  | (4.382) | (3.604) | (2.345) | (2.708) | (3.040) | (0.830) |
| Observations | 2,840 | 2,840 | 2,840 | 2,840 | 2,840 | 2,840 |
| R-squared | 0.591 | 0.651 | 0.691 | 0.662 | 0.643 | 0.856 |
| No. of cities | 284 | 284 | 284 | 284 | 284 | 284 |

Note: TOR is the dependent variable.

*** p < 0.01

** p < 0.05

* p < 0.1.

## 5.3. Optimizing tourism resources as a competitive advantage

Tourism resources in abundance can provide a comparative advantage for regions focusing on tourism development, but the resource curse phenomenon necessitates policymakers considering economic risks. Tourist cities must diversify their economies to avoid the 'resource curse.' Tourism resources can help mitigate the impact of air pollution on tourism economies [14], which is especially important for cities with abundant tourism resources and mining industries, such as Shanxi in China. Pollution has decreased since the coal industry capacity was reduced in 2015, but it has also decreased at a cost. On the other hand, Shanxi's rich historical and cultural resources present opportunities for high-quality tourism development,

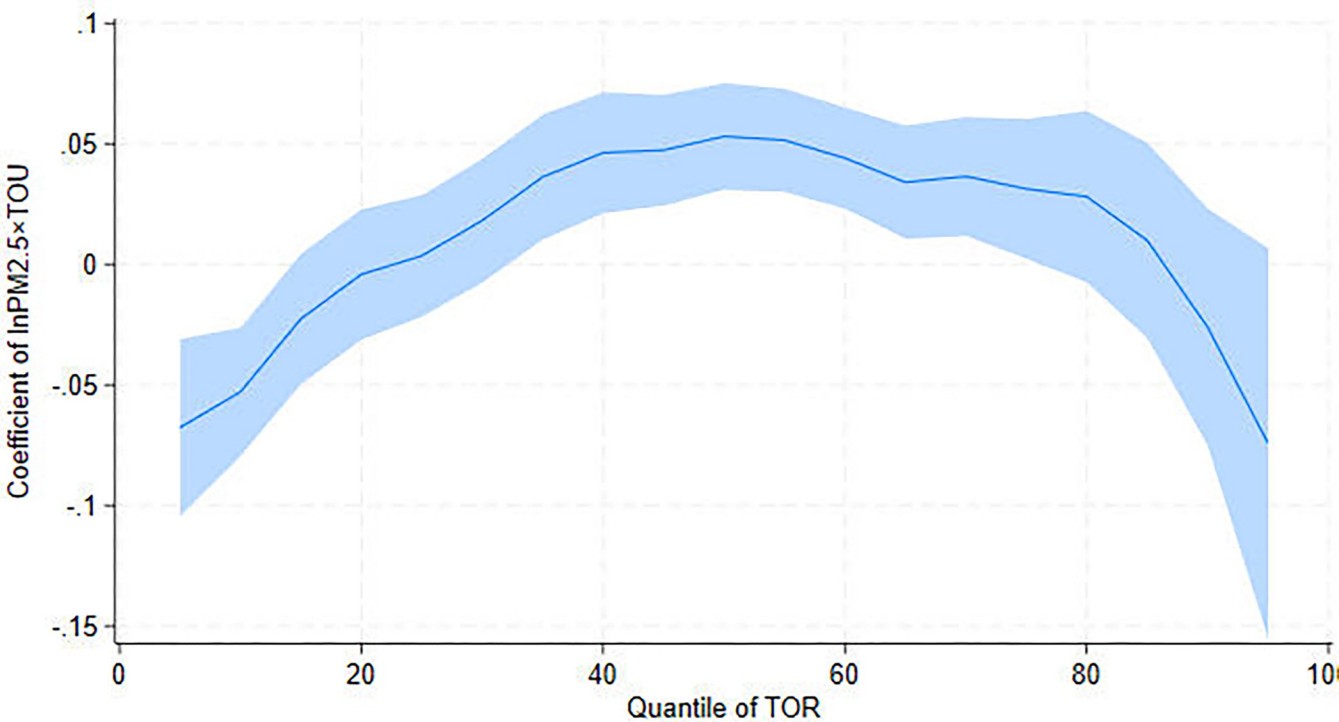

**Fig 6. UQR coefficient distribution for the interaction effects between PM$_{2.5}$ and tourism resource (TOU) on tourism revenue (TOR).**

transforming her resource-based economies. Improving tourism development resilience and risk resistance will be critical for future tourism development, and leveraging tourism prosperity gains for economic diversification can help.

## 6. Conclusions

The primary objective of this study is to examine the heterogeneous impact of air pollution on tourism across the cities in China. We have gathered extensive longitudinal panel data pertaining to the field of tourism. Our study employs UQR to account for the heterogeneity of observed individuals and distributions at various stages of tourism development. Furthermore, we explore the interaction effects of air pollution and tourism resources on tourism. Our statistical results reveal the impact is varied and diverse. Air pollution positively influences tourism (regarding tourist visits and tourism revenue) in areas with low tourism development levels, particularly in regions with inadequate infrastructure. However, a complex correlation between air pollution and tourism emerges when tourism development has reached a certain level. Briefly, the correlation is initially negative, then positive, and finally disappears. But, the overall correlation remains negative. The existing literature primarily supports the view that air pollution hurts tourism. While our findings do not reject this viewpoint, we highlight the variation of the effects among cities, emphasizing the need for a more place-based approach in tourism development strategies to take into account environmental factors. Besides, the effects of the interaction between air pollution and tourism resources on tourism are inverted U-shaped, implying that tourism resources can mitigate the negative effects of air pollution on tourism only when tourism development has reached a certain level.

In addition to these findings, our research comprehensively explores the impact of key variables in tourism, including transportation networks, industrial structure, population, and

**Table 8. UQR Q50 estimation results for PM$_{2.5}$ on tourism revenue (TOR) and tourist visits (TOV).**

| | (1) | (2) | (3) | (4) | (5) | (6) |
|---|---|---|---|---|---|---|
| | TOR | TOV | TOR | TOV | TOR | TOV |
| VARIABLES | Q50 | Q50 | Q50 | Q50 | Q50 | Q50 |
| ln$PM_{2.5}$ | -1.595*** | -0.951*** | -2.817 | -1.539 | -1.622*** | -1.284*** |
| | (0.248) | (0.217) | (2.775) | (2.211) | (0.294) | (0.255) |
| ln$PM_{2.5}^2$ | | | 0.158 | 0.076 | | |
| | | | (0.359) | (0.287) | | |
| ln$PM_{2.5}$*year | | | | | 0.006 | 0.074** |
| | | | | | (0.033) | (0.031) |
| ln$TOU$ | -0.016 | -0.012 | -0.016 | -0.012 | -0.016 | -0.014* |
| | (0.010) | (0.008) | (0.010) | (0.008) | (0.010) | (0.008) |
| ln$ROAD$ | -0.338* | -0.244 | -0.337* | -0.244 | -0.332 | -0.173 |
| | (0.201) | (0.197) | (0.201) | (0.198) | (0.202) | (0.187) |
| AIR | -0.054 | -0.029 | -0.054 | -0.030 | -0.053 | -0.021 |
| | (0.036) | (0.027) | (0.035) | (0.027) | (0.036) | (0.028) |
| RAIL | -0.057*** | -0.053*** | -0.057*** | -0.053*** | -0.057*** | -0.055*** |
| | (0.014) | (0.011) | (0.014) | (0.011) | (0.014) | (0.011) |
| D_ln$GDP$ | -0.045 | 0.007 | -0.040 | 0.009 | -0.047 | -0.014 |
| | (0.109) | (0.095) | (0.110) | (0.096) | (0.109) | (0.093) |
| D_TERIND | -0.001 | -0.000 | -0.000 | -0.000 | -0.001 | -0.000 |
| | (0.006) | (0.005) | (0.006) | (0.005) | (0.006) | (0.005) |
| ln$POP$ | 0.242 | -0.040 | 0.242 | -0.040 | 0.241 | -0.053 |
| | (0.162) | (0.140) | (0.160) | (0.140) | (0.162) | (0.139) |
| FISCAL | -0.092 | -0.133 | -0.084 | -0.130 | -0.088 | -0.090 |
| | (0.225) | (0.238) | (0.228) | (0.238) | (0.224) | (0.228) |
| TEMP | -0.062 | 0.003 | -0.058 | 0.005 | -0.064 | -0.024 |
| | (0.058) | (0.052) | (0.058) | (0.052) | (0.058) | (0.053) |
| WIND | -0.014 | 0.078 | -0.005 | 0.082 | -0.017 | 0.043 |
| | (0.096) | (0.082) | (0.096) | (0.083) | (0.097) | (0.081) |
| PREC | 0.662 | 0.765 | 0.674 | 0.771 | 0.660 | 0.743 |
| | (0.805) | (0.740) | (0.805) | (0.741) | (0.805) | (0.728) |
| Constant | 15.368*** | 14.419*** | 17.636*** | 15.510*** | 15.339*** | 14.056*** |
| | (2.344) | (2.273) | (5.557) | (4.646) | (2.341) | (2.171) |
| Observations | 2,840 | 2,840 | 2,840 | 2,840 | 2,840 | 2,840 |
| R-squared | 0.688 | 0.683 | 0.688 | 0.683 | 0.688 | 0.685 |
| No. of cities | 284 | 284 | 284 | 284 | 284 | 284 |

Note: TOR Q50 / TOV Q50 is the dependent variable

*** p < 0.01

** p < 0.05

* p < 0.1.

climate. Based on our findings, we recommend that policymakers prioritize air quality to achieve sustainable tourism development. Depending on the city's tourism development level, lower pollution levels can be achieved through technological innovations in tourism resources, the establishment of comprehensive transportation systems, and diversified industrial structures. By implementing the above measures, cities can foster sustainable tourism growth while mitigating the negative impacts of air pollution on tourism.

Our study on the heterogeneous impact of air pollution on tourism recognizes limitations in both data and methodology. The indicators used in our study may not capture all aspects of industry progression. To minimize the influence of Covid-19, we focused on the pre-2019 data. For future research, we propose examining periods during and after pandemics, investigating the impact of Covid-19 on tourism and the tourism industry's adaptive strategies. Additionally, we recommend including additional variables, such as policies, economies, cultural dynamics, and climate change, in statistical models for a more holistic understanding of tourism development trends and challenges.

## Supporting information

**S1 File. The data employed in this study.**
(XLSX)

## Acknowledgments

We thank Antony Andrews and another reviewer for their constructive feedback and suggestions on this manuscript.

## Author Contributions

**Conceptualization:** Yuxuan Xiao, Will W. Qiang, Harry F. Lee.

**Data curation:** Yuxuan Xiao, Will W. Qiang, Harry F. Lee.

**Formal analysis:** Yuxuan Xiao, Will W. Qiang, Harry F. Lee.

**Investigation:** Yuxuan Xiao, Will W. Qiang, Harry F. Lee.

**Methodology:** Yuxuan Xiao, Will W. Qiang, Harry F. Lee.

**Validation:** Yuxuan Xiao, Will W. Qiang, Chung-Shing Chan, Steve H. L. Yim, Harry F. Lee.

**Writing – original draft:** Yuxuan Xiao, Will W. Qiang, Chung-Shing Chan, Steve H. L. Yim, Harry F. Lee.

**Writing – review & editing:** Yuxuan Xiao, Will W. Qiang, Chung-Shing Chan, Steve H. L. Yim, Harry F. Lee.

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
