## [Decision Letter · Decision Letter 0]

1 Dec 2023

PONE-D-23-30994How Much Can Air Pollution Influence Tourism? Examination of 284 Chinese Prefecture-level CitiesPLOS ONE

Dear Dr. Lee,

Thank you for submitting your manuscript to PLOS ONE. After careful consideration, we feel that it has merit but does not fully meet PLOS ONE’s publication criteria as it currently stands. Therefore, we invite you to submit a revised version of the manuscript that addresses the points raised during the review process.

Please submit your revised manuscript by Jan 15 2024 11:59PM. If you will need more time than this to complete your revisions, please reply to this message or contact the journal office at plosone@plos.org. Please include the following items when submitting your revised manuscript:A rebuttal letter that responds to each point raised by the academic editor and reviewer(s). You should upload this letter as a separate file labeled 'Response to Reviewers'.A marked-up copy of your manuscript that highlights changes made to the original version. You should upload this as a separate file labeled 'Revised Manuscript with Track Changes'.An unmarked version of your revised paper without tracked changes. You should upload this as a separate file labeled 'Manuscript'.

We look forward to receiving your revised manuscript.

Kind regards,

Nikeel Nishkar Kumar

Academic Editor

PLOS ONE

Reviewers' comments:

Reviewer's Responses to Questions

**Comments to the Author**

1. Is the manuscript technically sound, and do the data support the conclusions?

Reviewer #1: Partly

Reviewer #2: Yes

2. Has the statistical analysis been performed appropriately and rigorously? 

Reviewer #1: Yes

Reviewer #2: I Don't Know

3. Have the authors made all data underlying the findings in their manuscript fully available?

Reviewer #1: No

Reviewer #2: Yes

4. Is the manuscript presented in an intelligible fashion and written in standard English?

Reviewer #1: Yes

Reviewer #2: Yes

5. Review Comments to the Author

Reviewer #1: How Much Can Air Pollution Influence Tourism? Examination of 284 Chinese Prefecture-level Cities

Reviewer report:

Congratulations on the comprehensive work presented in this manuscript. However, before this can make it to press, I have several questions and suggestions that require clarification:

1. How was the weighting system for tourism resources (e.g., world cultural heritage, national scenic spots) determined, and what was the rationale behind assigning specific weights to each category?

2. Given that there are multiple sources to obtain PM2.5 concentration data, such as direct information from China’s environmental bureau and the LandScan method, how do you ensure consistency and accuracy in the PM2.5 measurements used in this study?

3. The study references the use of the artificial impervious area dataset by Gong et al. (2020) for specifying the spatial coverage of urban areas. How does this dataset compare with other available methods, such as the GHS build-up grids, in terms of capturing the true essence of urban built-up areas in China?

4. With respect to the transportation infrastructure variables, how did you handle cities that might have multiple railway stations or airports of different grades? For instance, if a city has both a third and fifth-level railway station, how is that incorporated into the model?

5. The study controls for various socioeconomic factors like GDP and the ratio of public fiscal expenditure to GDP. Are there any potential confounding variables that were not included in the model, and how might they influence the findings?

Empirical Method:

6.

• The method assumes observations are independent, which can be problematic for time-series or spatial data.

• With multiple coefficients for each quantile, the interpretation and comparison of results can be intricate.

• Have you considered simpler alternatives, such as Median Regression? This approach targets the median (or 50th quantile) of the dependent variable and offers a robust analysis without assuming specific error distributions. Not only is it less computationally intensive, but it also provides a more parsimonious view of the data, focusing on the median's relationships.

Could you kindly elaborate on your choice of QR over such alternatives and address the aforementioned concerns?

7. Your analysis mentioned that in cities with low tourist visit numbers (TOV), higher PM2.5 is associated with an increase in TOV, which contradicts the findings of Shah et al. (2022). Could you elaborate on the potential factors or unique circumstances present in these cities that might account for this positive correlation between air pollution and tourism? Additionally, how do you reconcile this with the prevailing literature that often associates increased air pollution with a reduction in tourism?

8. Given that the interaction effect between PM2.5 and TOU on tourist visits (TOV) is non-linear, how does the magnitude of this effect vary across different percentiles (Q10, Q50, and Q75) and what implications does this have for tourism development in regions with varying air pollution levels?

9. The study highlights the potential dangers of the "resource curse" in the context of tourism development. How can cities with abundant tourism resources, such as Shanxi, ensure they capitalize on their historical and cultural assets without falling victim to the pitfalls of over-reliance on a single industry

Reviewer #2: - This is a interesting paper.

- Gap and Contribution: The paper needs to improve the indentification of the gap and its contribution. The arguement used for a cotrasting view is based on two different context of papers. One is based on people's perception and the other uses secondary data. Therefore to make the arguement, both contrasting views should be based on the same type of data.

- Policy implications based on findings: The findings are interesting and therefore the author(s) need to provide appropriate and specific policy implications based on this research.

- Please follow the journal's formatting guidelines.

- All the best.

6. PLOS authors have the option to publish the peer review history of their article (what does this mean?). If published, this will include your full peer review and any attached files.

Reviewer #1: **Yes: **Antony Andrews

Reviewer #2: No

---

## [Author Response · Author response to Decision Letter 0]

5 Jan 2024

We have uploaded the "Response to Reviewers" in this submission. Thanks!

---

## [Decision Letter · Decision Letter 1]

7 Mar 2024

PONE-D-23-30994R1How Much Can Air Pollution Influence Tourism? Examination of 284 Chinese Prefecture-level CitiesPLOS ONE

Dear Dr. Lee,

Thank you for submitting your manuscript to PLOS ONE. After careful consideration, we feel that it has merit but does not fully meet PLOS ONE’s publication criteria as it currently stands. Therefore, we invite you to submit a revised version of the manuscript that addresses the points raised during the review process.

We look forward to receiving your revised manuscript.

Kind regards,

Nikeel Nishkar Kumar

Academic Editor

PLOS ONE

Reviewers' comments:

Reviewer's Responses to Questions

**Comments to the Author**

1. If the authors have adequately addressed your comments raised in a previous round of review and you feel that this manuscript is now acceptable for publication, you may indicate that here to bypass the “Comments to the Author” section, enter your conflict of interest statement in the “Confidential to Editor” section, and submit your "Accept" recommendation.

Reviewer #1: All comments have been addressed

2. Is the manuscript technically sound, and do the data support the conclusions?

Reviewer #1: Partly

3. Has the statistical analysis been performed appropriately and rigorously? 

Reviewer #1: I Don't Know

4. Have the authors made all data underlying the findings in their manuscript fully available?

Reviewer #1: Yes

5. Is the manuscript presented in an intelligible fashion and written in standard English?

Reviewer #1: Yes

6. Review Comments to the Author

Reviewer #1: How Much Can Air Pollution Influence Tourism? Examination of 284 Chinese Prefecture-level Cities

Thanks for responding to my quieries – I am generally satisfied, just have couple of points to address before it cn make it to press:

1. Authors Response: Thank you for the question. Yes, factors such as weather conditions,

wind speed, and Air Quality Index (AQI) could potentially influence our findings.

These factors have been shown in prior studies to impact air pollution and tourism

development in individual cities or smaller samples. Regrettably, acquiring such

granular data is difficult in large-scale studies. Hence, we haven’t incorporated these

factors into this study.

Reviewer’s Response: Thank you for your response. I appreciate your acknowledgment of the potential influence of external factors like weather conditions, wind speed, and Air Quality Index (AQI) on your study's findings. It's understandable that acquiring granular data for such variables can be challenging in large-scale studies. Your decision to exclude these factors seems reasonable given the logistical difficulties in obtaining this detailed information.

However, it might be beneficial to discuss these limitations in your paper. Addressing the absence of these variables could provide readers with a clearer understanding of the study's scope and the context in which your findings are situated. This would also help in setting future research directions where such data might be more readily available or in smaller scale studies where the impact of these factors can be more feasibly assessed.

2. My other question is about your model itself, and I am sorry to bring this up during the second review, but it is important for the credibility of your findings:

In the context of examining the relationship between air pollution and tourism development across a substantial number of cities over a decade, I have some concerns and questions regarding the handling of potential non-stationarity and trend components in your data:

Accounting for Linear Trends: The data spans 284 cities over a period from 2008 to 2018, which is a substantial time frame. However, it appears that linear trends have not been explicitly accounted for in your model. Could you elaborate on how your model addresses or accommodates potential linear trends in both the tourism development and air pollution variables over this period?

Potential Non-Stationarity Issues: Given the longitudinal nature of your data, there is a possibility of non-stationarity within the panel dataset. Non-stationarity can lead to biased and inconsistent coefficient estimates. How do you ensure that the variables in your panel data are stationary, or how do you mitigate the effects of potential non-stationarity?

Implications of Omitting Trend Components: If linear trends are not adequately accounted for, this might lead to spurious regression problems. Can you discuss the potential implications of this omission on the interpretation of your results, particularly regarding the coefficients' magnitude and significance?

Strategies for Trend and Non-Stationarity Adjustment: Have you considered using detrending methods or incorporating a trend variable in your model? Additionally, are there other econometric techniques, such as first differencing or cointegration analysis, that you have employed or could employ to address these issues?

You'll likely need to revise several models. A practical starting point would be conducting unit root tests specifically designed for panel data. For instance, using Stata, you can perform the Levin-Lin-Chu (LLC) test, Im-Pesaran-Shin (IPS) test, and Fisher-type tests based on augmented Dickey-Fuller (ADF) tests. If you prefer EViews, it also supports tests like the LLC, IPS, and Maddala and Wu's Fisher-type tests for such analyses. Moreover, the R programming language offers robust packages like plm and urca for panel data analysis, which include unit root tests such as the LLC and IPS tests.

I recommend applying these unit root tests to all your OLS models. Ideally, we won't find any unit roots, as their presence could complicate matters, potentially requiring significant revisions. Even if the tests for unit roots are negative, consider including a trend variable in your models, possibly both linear and quadratic trends, to enhance the robustness of your analysis.

Good luck with revision

7. PLOS authors have the option to publish the peer review history of their article (what does this mean?). If published, this will include your full peer review and any attached files.

Reviewer #1: **Yes: **Antony Andrews

---

## [Author Response · Author response to Decision Letter 1]

12 Apr 2024

Our response to reviewer has been attached to this submission. Thanks!

---

## [Decision Letter · Decision Letter 2]

18 Apr 2024

PONE-D-23-30994R2How Far Can Air Pollution Affect Tourism in China? Evidence from Panel Unconditional Quantile RegressionsPLOS ONE

Dear Dr. Lee,

Thank you for submitting your manuscript to PLOS ONE. After careful consideration, we feel that it has merit but does not fully meet PLOS ONE’s publication criteria as it currently stands. Therefore, we invite you to submit a revised version of the manuscript that addresses the points raised during the review process.

We look forward to receiving your revised manuscript.

Kind regards,

Nikeel Nishkar Kumar

Academic Editor

PLOS ONE

Journal Requirements:

Reviewers' comments:

Reviewer's Responses to Questions

**Comments to the Author**

1. If the authors have adequately addressed your comments raised in a previous round of review and you feel that this manuscript is now acceptable for publication, you may indicate that here to bypass the “Comments to the Author” section, enter your conflict of interest statement in the “Confidential to Editor” section, and submit your "Accept" recommendation.

Reviewer #1: (No Response)

2. Is the manuscript technically sound, and do the data support the conclusions?

Reviewer #1: Partly

3. Has the statistical analysis been performed appropriately and rigorously? 

Reviewer #1: No

4. Have the authors made all data underlying the findings in their manuscript fully available?

Reviewer #1: Yes

5. Is the manuscript presented in an intelligible fashion and written in standard English?

Reviewer #1: Yes

6. Review Comments to the Author

Reviewer #1: Good work. But believe it's essential to include a trend component in the regression model. Despite the authors having conducted unit root tests, my previous comment underscored the importance of incorporating trend variables, possibly both linear and quadratic, to strengthen the robustness of your analysis. It would be advisable for the authors to present two sets of results: one that includes the trend component and another without it. We want to make sure that your coefficients do not change very much. I strongly believe that the authors should include a trend component in their model to derive the coefficient, which could provide insights into technological progress or total factor productivity, depending on their interpretation. Good luck :)

7. PLOS authors have the option to publish the peer review history of their article (what does this mean?). If published, this will include your full peer review and any attached files.

Reviewer #1: **Yes: **Antony Andrews

---

## [Author Response · Author response to Decision Letter 2]

5 May 2024

Our response to reviewer has been attached to this submission. Thanks!

---

## [Editor Report · Decision Letter 3]

10 May 2024

How Far Can Air Pollution Affect Tourism in China? Evidence from Panel Unconditional Quantile Regressions

PONE-D-23-30994R3

Dear Dr. Lee,

We’re pleased to inform you that your manuscript has been judged scientifically suitable for publication and will be formally accepted for publication once it meets all outstanding technical requirements.

Kind regards,

Nikeel Nishkar Kumar

Academic Editor

PLOS ONE

---

## [Editor Report · Acceptance letter]

14 May 2024

PONE-D-23-30994R3 

PLOS ONE

Dear Dr. Lee, 

I'm pleased to inform you that your manuscript has been deemed suitable for publication in PLOS ONE. Congratulations! Your manuscript is now being handed over to our production team.

Kind regards, 

on behalf of

Dr. Nikeel Nishkar Kumar 

Academic Editor

PLOS ONE